# Targeted Palliative Radionuclide Therapy for Metastatic Bone Pain

**DOI:** 10.3390/jcm9082622

**Published:** 2020-08-12

**Authors:** Reyhaneh Manafi-Farid, Fardad Masoumi, Ghasemali Divband, Bahare Saidi, Bahar Ataeinia, Fabian Hertel, Gregor Schweighofer-Zwink, Agnieszka Morgenroth, Mohsen Beheshti

**Affiliations:** 1Research Center for Nuclear Medicine, Shariati Hospital, Tehran University of Medical Sciences, Tehran 1411713135, Iran; manafi_farid_r@hotmail.com (R.M.-F.); bahare_saidi@yahoo.com (B.S.); 2Department of Orthopedic and Trauma Surgery, Shariati Hospital, Tehran University of Medical Sciences, Tehran 1411713135, Iran; masoumi.fardad@gmail.com; 3Department of Nuclear Medicine, Jam Hospital, Tehran 1588657915, Iran; divband_ali@yahoo.com; 4Athinoula A Martinos Center for Biomedical Imaging, Department of Radiology, Massachusetts General Hospital—Charlestown HealyhCare Center, Boston, MA 02129, USA; bahar.ataeinia@gmail.com; 5Department of Nuclear Medicine, University Hospital, RWTH University, 52074 Aachen, Germany; fhertel@ukaachen.de (F.H.); amorgenroth@ukaachen.de (A.M.); 6Department of Nuclear Medicine and Endocrinology, Paracelsus Medical University, 5020 Salzburg, Austria; g.schweighofer-zwink@salk.at

**Keywords:** bone-pain palliation, radionuclide therapy, bone-seeking, [^89^Sr]strontium, [^153^Sm]Sm–EDTMP, [^223^Ra]RaCl, [^188^Re]Re–HEDP, [^186^Re]Re–HEDP, [^177^Lu]lutetium-EDTMP, combination therapy

## Abstract

Bone metastasis develops in multiple malignancies with a wide range of incidence. The presence of multiple bone metastases, leading to a multitude of complications and poorer prognosis. The corresponding refractory bone pain is still a challenging issue managed through multidisciplinary approaches to enhance the quality of life. Radiopharmaceuticals are mainly used in the latest courses of the disease. Bone-pain palliation with easy-to-administer radionuclides offers advantages, including simultaneous treatment of multiple metastatic foci, the repeatability and also the combination with other therapies. Several β¯- and α-emitters as well as pharmaceuticals, from the very first [^89^Sr]strontium-dichloride to recently introduced [^223^Ra]radium-dichloride, are investigated to identify an optimum agent. In addition, the combination of bone-seeking radiopharmaceuticals with chemotherapy or radiotherapy has been employed to enhance the outcome. Radiopharmaceuticals demonstrate an acceptable response rate in pain relief. Nevertheless, survival benefits have been documented in only a limited number of studies. In this review, we provide an overview of bone-seeking radiopharmaceuticals used for bone-pain palliation, their effectiveness and toxicity, as well as the results of the combination with other therapies. Bone-pain palliation with radiopharmaceuticals has been employed for eight decades. However, there are still new aspects yet to be established.

## 1. Introduction

Bone metastasis is a common feature of malignancies, particularly in late stages [1,2]. The incidence varies widely in different tumors [1,2]. Up to 70–80% of the cases reportedly occur in prostate, breast and lung cancers [3,4]. The presence of bone metastases implies poorer prognosis, shortens survival and is associated with a multitude of complications, including severe bone pain, pathological fracture, spinal cord compression, hypercalcemia, etc. [3,5]. Bone metastases are found using various imaging modalities, namely, plane radiography, computed tomography (CT) and magnetic resonance imaging (MRI), as well as functional examinations employing a wide range of targeted radioligands with single photon emission tomography (SPECT), positron emission tomography (PET) and hybrid SPECT/CT and PET/CT imaging [6,7].

One of the disabling presentations of bone metastases is refractory bone pain. The issue to enhance the quality of life of the patients in their remaining life span continues to be challenging. Bone pain is managed through multidisciplinary approaches using analgesics, bisphosphonates, chemotherapy, external beam radiotherapy (EBRT), immunotherapy, surgery, hormonal treatments and finally, bone-targeted radionuclide therapy [8,9,10].

Bone-pain palliation with easy-to-administer radionuclides offers advantages including simultaneous treatment of multiple metastatic foci, the repeatability and also the combination with other treatments [3,10]. In a systematic review including 57 studies, the response rate was reported 70% using β¯-emitters [11]. Not only does radionuclide targeted therapy alleviate pain, but it reduces or defers the incidence of skeletal-related events (SRE) [10]. These agents substitute calcium or bind to hydroxyapatite in bones and deliver ionizing radiation to areas with increased osteoblastic activity [12,13]. It is of substantial importance to deliver the utmost radiation to metastatic foci while sparing non-affected tissues. Hence, numerous radiopharmaceuticals have been investigated to achieve the optimum result. The extent of metastatic disease, renal function, bone marrow reserve, and, importantly, the availability impact the physicians’ choice of an appropriate radiopharmaceutical [3]. A slightly higher response rate has been reported in patients in whom lesions are osteoblastic [13], the skeletal involvement is limited and the performance state is higher [14].

Bone-seeking radiopharmaceuticals emitting β¯-particles have long been used for bone-pain palliation. Meanwhile, α-emitting tracers, with [^223^Ra]radium-dichloride at the top, were also developed. Alpha-emitters provide more toxicity for tumoral cells and lesser radiation for surrounding normal tissues [12]. They dispense higher linear energy in a shorter range (<100 μm) and predominantly induce permanent DNA double-strand breaks [12]. In addition to bone-seeking agents, other targeted radiopharmaceuticals are used for systemic therapy in nuclear medicine showing therapeutic effects on bone metastases in prostate and neuroendocrine tumors [15,16].

In this review, we aimed to provide an overview of a number of long-established and novel bone-seeking radiopharmaceuticals used for bone-pain palliation, including [^89^Sr]strontium-dichloride, [^153^Sm]samarium, [^186^Re]rhenium, [^188^Re]rhenium, [^177^Lu]lutetium and [^166^Ho]holmium labeled with bone-seeking agents, as well as [^223^Ra]radium-dichloride. Noteworthy, [^32^P]Phosphorus-orthophosphate and [^117m^Sn]-diethylenetriamine pentaacetic acid have traditionally been used; however, their application for bone-pain palliation is currently limited and are not discussed [17,18]. Moreover, the effectiveness and toxicity of the radiopharmaceuticals, as well as the results of the combination therapy with other therapies are also briefly mentioned (Table 1).

## 2. [89. Sr]Strontium-Dichloride ([^89^Sr]SrCl)

[^89^Sr]SrCl is used for bone-pain palliation from 1942 by Pecher [39]. Its efficacy is well-documented in the literature [14,39].

[^89^Sr]strontium mimics calcium in the body and is taken up into the inorganic bone matrix [8]. Its concentration in the skeleton is a proportion of osteoblastic activity, which is 10-fold higher in metastatic foci [39]. After localization, it remains still in tumoral sites for 100 days [39]. The excretion occurs predominantly from kidneys [19], limiting its use in the setting of renal failure.

According to guidelines, 150 MBq [^89^Sr]SrCl is administered with slow intravenous infusion [3]. The response rate has been documented to be between 60 to 95% [40]. In the latest meta-analysis, the overall response rate of 70% has been reported [11], commencing from the 4–28 days (typically in 14–28 days) of administration and lasting up to 15 months (typically 12–26 weeks) [9,20]. A flushing sensation may occur in case of rapid infusion [3]. The hematological toxicity is the major side-effect of [^89^Sr]SrCl which is mild, temporary and predominantly consists of myelosuppression [9,41]. The nadir occurs between the 12–16 weeks [3], showing resolution in the next six weeks depending on the skeletal tumor extent and bone marrow reserve [20,41]. Follow-up of hematological toxicity is recommended for 12–16 weeks, for the prolonged effect on bone marrow [3].

Despite an acceptable response rate with [^89^Sr]SrCl, a recent meta-analysis showed no significant benefit in prolonging overall survival (OS) or symptomatic SRE-free survival in metastatic castration-resistant prostate cancer (mCRPC) [42]. Besides pain reduction, significant improvement of quality of life has been reported [43,44]. However, the improvement in quality of life generally follows pain reduction [45].

Moreover, the combination of [^89^Sr]SrCl with other therapies has been investigated, mainly in prostate cancer showing the overall response rate of 74% [11]. Nevertheless, the data considering the currently used agent in mCRPC patients (docetaxel) is scarce, showing improved clinical progression-free survival (PFS), but no effect on OS or SRE-free survival [46].

[^89^Sr]SrCl and EBRT reportedly appear to perform rather similar in reducing pain and disease progression [47]. In one survey, EBRT slightly outperformed [^89^Sr]SrCl in terms of OS [47]. In combination, however, the results are controversial. The pain relief was higher in patients receiving both [^89^Sr]SrCl and EBRT compared to [^89^Sr]SrCl-alone in one study [48], while there was no impact of the addition of [^89^Sr]SrCl in another [49].

Finally, the combination therapy of [^89^Sr]SrCl and zoledronic acid for bone metastases has shown superiority in terms of reduction of bone pain, analgesic drug use and time to decrease in pain, as well as improvement of the quality of life compared to [^89^Sr]SrCl- or zoledronic acid-alone [50]. There has been no significant higher rate of toxicity in the combined method [50]. Moreover, using the combination therapy, pain relief has been reported in 94% of the patients, tumoricidal effect in 36%, and non-progressive disease in 86% [51]. Additionally, a significant increase in OS and time to first SRE have been noted with the combination compared to no treatment [52]. In addition, the combination of [^89^Sr]SrCl and zoledronic acid has revealed survival benefits compared to [^89^Sr]SrCl-alone [53].

Of note, the efficacy of [^89^Sr]SrCl in bone-pain palliation has been compared to other radiopharmaceuticals revealing no significant difference to [^153^Sm]samarium-ethylene diamine tetramethylene phosphonate and [^186^Re/^188^Re]rhenium-hydroxyethylidene diphosphonate [11,14,18,54]. [^89^Sr]SrCl is still an effective method for bone-pain palliation. The combination with other treatments seem more appealing warranting further investigations to precisely evaluate the impact of these protocols.

## 3. [153. Sm]Samarium–Ethylene Diamine Tetramethylene Phosphonate ([^153^Sm]Sm–EDTMP)

[^153^Sm]Sm–EDTMP, a well-known radiopharmaceutical for bone-pain palliation, has been widely used since FDA approval in various osteoblastic metastatic lesions, especially in prostate and breast cancer [55,56,57]. It rapidly binds to hydroxyapatite crystals, leading to less than 1% availability in the blood 5 h after injection. The excretion occurs mainly through the kidneys [58,59].

Although the maximum tolerable dose of 111 MBq/kg of body weight has been reported in initial dosimetry studies [58,60], the standard dose of 37 MBq/kg is well-established [3]. In numerous affirmative studies, a response rate of 40–97% [22,23,61,62,63] with a mean of 70% [11] has been reported. In addition, a decrease in analgesic consumption and improved quality of life has been noted [22,23,24]. The pain reduction occurs as early as in the first week after injection [23,61], lasting for about two to three months [23,24,25].

The bone marrow toxicity is mild in most the patients [61,64]. The lowest platelet and white blood cell (WBC) counts have been observed in 3–5 weeks, recovering in 6–8 weeks after therapy [23,24,62,64]. Patients with severe grades of myelotoxicity had predisposing underlying conditions including recent chemotherapy, EBRT or malignant bone marrow involvement [65]. Similarly, mainly minimal and transient bone marrow toxicity has been observed after administering repeated doses of [^153^Sm]Sm–EDTMP [66,67]. Other complications are rare with [^153^Sm]Sm–EDTMP compared to systemic therapies like chemotherapy [55].

Moreover, the impact of combining [^153^Sm]Sm–EDTMP with other therapies has been evaluated in a few studies. Adding [^153^Sm]Sm–EDTMP to docetaxel for mCRPC patients has revealed favorable results regarding pain control, safety and a decrease in PSA level [68,69,70]. The PFS has been reported 5.2 to 7 months [68,69,70].

In addition, combined therapy with local EBRT has resulted in higher analgesic effect than monotherapy with [^153^Sm]Sm–EDTMP, showing no significant difference in bone marrow toxicity in mCRPC [71]. Similar outcomes were reported for combination therapy with bisphosphonates such as zoledronic acid, leading to significantly better pain control [72,73], higher quality of life [73] and shorter time to analgesia [72].

Overall, [^153^Sm]Sm–EDTMP has been successfully used for pain control in more than three decades. It seems the combination with other therapies results in a better outcome. However, the studies are not sufficient to suggest a definite conclusion. Also, earlier initiation of [^153^Sm]Sm–EDTMP may be more effective [74]. Hence, the optimal timing of administration of [^153^Sm]Sm–EDTMP requires further assessment.

## 4. [186. Re]Rhenium-Hydroxyethylidene Diphosphonate ([^186^Re]Re–HEDP)

[^186^Re]rhenium is a combined β¯/γ-emitter isotope labelled with a phosphonate complex (HEDP), localized in bones attaching hydroxyapatite crystals [17]. The shorter physical half-life of 3.8 days may produce a faster onset of pain relief in comparison to the radiopharmaceuticals with longer half-lives [75]. The usual recommended dose of [^186^Re]Re–HEDP for bone-pain palliation therapy is 1295 MBq (35 mCi) [26]. Approximately 70% of the administrated activity is excreted in the urine within the first 24 h after injection [17].

The successful pain reduction has been noted in different studies using [^186^Re]Re–HEDP [76,77,78]. However, the large cohorts addressing [^186^Re]Re–HEDP is lacking. The overall response rate has been reported to be between 38% and 82% using doses of 1295 MBq to 1406 MBq [17]. While it has been mainly used for breast and prostate cancers, limited studies in other malignancies, such as lung cancer, have revealed similar performance [78,79,80]. Additionally, improved quality of life and reduction of analgesic medication has been reported [79,81]. Usually, the response occurs in 1–3 weeks after injection [76,82] and lasts for 3 weeks to 12–15 months [76,81,82,83]. After therapy, a temporary decrease in platelets and WBC counts frequently occurs [76,78,83], reaches the nadir after 1–5 weeks [26,83] and recovers in the following 2–4 weeks [83,84].

Although survival benefits have not been proved for β¯-emitters, a study on 57 mCRPC patients, high-dose treatment (>3500 MBq) has shown an increase in OS, in the univariate analysis. However, in the multivariate analysis, only the lower disease burden was associated with increased OS [85]. There is literally little studies evaluating the combination with other therapies. However, the safety in co-administration with docetaxel and zoledronic acid has been documented [86,87].

No significant difference in rates of pain palliation, therapeutic efficacy or the possibility of toxicity has been reported for [^186^Re]Re–HEDP compare to other β¯-emitters, including [^188^Re]Re–HEDP, [^153^Sm]Sm-EDTMP and [^89^Sr]SrCl [54,88]. Hence, [^186^Re]Re–HEDP can be considered as a reasonable choice for bone-pain palliation therapy regarding logistic feasibility. Further studies are required to establish its role in combination with other treatments.

## 5. [188. Re]Rhenium-Hydroxyethylidene Diphosphonate ([^188^Re]Re-HEDP)

[^188^Re]Re–HEDP is a bone-seeking therapeutic radiopharmaceutical with favorable characteristics for application in bone-pain palliation therapy [89]. In clinical studies with [^188^Re]Re–HEDP, various dosage have been used, mainly between 1300–4400 MBq [27,28,90]. The maximum tolerated dose of 3300 MBq has been proposed in one study [29]. Approximately 40% of the administered activity is cleared within 8 h after injection, via the urinary system, resulting in a low radiation-dose to the whole-body [30]. Approximately 40% is localized in the skeleton at 24 h [90]. The localization in the bone metastases, however, is rather prolonged (half-life of 269 ± 166 h in bone metastases versus short whole-body biologic half-life of 51 ± 43 h) [30].

The response rate of 70–80% has been reported in painful bone metastases in various malignancies [90,91,92,93,94]. In addition, an increase in Karnofsky performance score and improved quality of life have been noticed after treatment with [^188^Re]Re–HEDP [28,93,94,95]. Generally, serious hematological side-effects are not expected [92,95,96]. The nadir for platelet and WBC counts occurs in 2–5 weeks and resolves in 8–12 weeks after therapy [28,54,94].

Although β¯-emitters are so-called “palliative”, repeated doses of [^188^Re]Re–HEDP, compared to a single administration, have shown improvement in PFS and OS [97], as well as a reduction in prostate-specific antigen (PSA) levels in approximately half of the patients [97,98]. Another retrospective study on sixty mCRPC patients has revealed that repeated treatment with [^188^Re]Re–HEDP can improve OS up to 15.6 months [99]. The apparent antitumoral effect may be explained by higher β¯-energy and tissue penetration, as well as a higher dose rate of [^188^Re]Re–HEDP.

Moreover, the efficacy of combination therapy has been investigated. Repeated doses of [^186^Re]Re–HEDP combined with docetaxel and prednisone versus docetaxel and prednisone alone have shown no significant improvement in PFS nor pain scores in 88 mCRPC patients [100]. In contrast, the combination with pamidronate in 48 breast cancer patients has revealed a better therapeutic effect, compared to either [^188^Re]Re–HEDP or pamidronate alone [101]. Furthermore, zoledronic acid has been labeled with [^188^Re]Re and has demonstrated promising results requiring confirmation in larger studies [102].

In conclusion, [^188^Re]Re–HEDP has not been approved in many countries for clinical use, and the number of prospective trials with large populations is limited. However, respecting physical characteristics, favorable pain control, the potential impact on OS, availability (on-site generator) and cost-effectiveness, it seems a proper radiopharmaceutical for bone-pain palliation and deserves further investigations.

## 6. [177. Lu]Lutetium-Ethylene Diamine Tetramethylene Phosphonate ([^177^Lu]Lu-EDTMP)

[^177^Lu]Lutetium labelled with somatostatin analogs and prostate-specific membrane antigen (PSMA) are generally used for the treatment of soft-tissue and bone metastases of neuroendocrine tumors and prostate cancer, respectively, revealing acceptable response rates [15,16]. Beside to this common application, [^177^Lu]Lu–EDTMP has been studied as a safe and effective potential palliative therapy in painful bone metastases, regarding binding to hydroxyapatite crystals, rapid skeletal accumulation and minimal uptake in other organs [103]. [^177^Lu]Lu–DOTMP (1,4,7,10-tetraazacyclododecane-1,4,7,10-tetramethylene phosphonate) has also been investigated revealing rather similar characteristics to [^177^Lu]Lu–EDTMP. Yet, the latter exhibits slightly higher skeletal uptake as well as retention in the liver and kidneys [33].

The combined complete and partial response rate of 77–100% has been documented in different studies using [^177^Lu]Lu–EDTMP for pain palliation of bone metastases [21,31,34]. The decrease in pain has been reported as early as 8–14 days after therapy lasting up to 12 weeks [31,34]. In addition, significant improvement in the quality of life has been noted in patients received [^177^Lu]Lu–EDTMP [21,31]. Comparing low- and high-dose therapy (1295 vs. 2590 MBq) in two studies, higher response rates have been reported in the high-dose group, although not statistically significant [21,31]. The hematological toxicity is the major complication which ends up with a decrease in peripheral blood cell count with a nadir in 4–8 weeks and gradual recovery in 12 weeks [21]. This side-effect is commonly insignificant [21,34]; However, transient grade III/IV hematotoxicity has been reported in 23% of patients [31].

Moreover, the effect of [^177^Lu]Lu–EDTMP has been compared to [^153^Sm]Sm–EDTMP. Reportedly, they both have subjected bone metastases to similar radiation doses [104]. Likewise, the response rate of approximately 75–80% has been noted for both radiopharmaceuticals [104,105,106]. In addition, the cocktail therapy of [^177^Lu]Lu-/[^153^Sm]Sm–EDTMP has shown safety in administration and pain relief/reduction in 24/25 patients [107].

[^177^Lu]Lu labeled with zoledronic acid ([^177^Lu]Lu-DOTA^ZOL^) is another investigational radiopharmaceutical with promising preliminary biodistribution and post-therapy dosimetry results. It also possess a potential theragnostic application (labeled with [^68^Ga]gallium–DOTA^ZOL^) for treatment of bone metastases [108].

To recapitulate, [^177^Lu]lutetium may be more available considering the established use in radionuclide therapy for neuroendocrine tumors and prostate cancers. Also, the features of relatively inexpensive [^177^Lu]lutetium, such as favorable half-life, production in high-specific activity, efficient therapeutic β¯-particles, longer effect with a single administration, lower radiation of bone marrow and sufficient γ-photons for imaging make it an interesting radioisotope in bone-pain palliation therapy [32,33,34,109,110]. The impact of [^177^Lu]Lu-phosphonate complexes in increasing survival indices is not addressed, fairly. More studies are required to investigate the clinical value of these radiopharmaceuticals, especially in combination with other therapies.

## 7. [166. Ho]holmium-Ethylene Diamine Tetramethylene Phosphonate ([^166^Ho]Ho-EDTMP)

Different phosphonate complexes have been labelled with [^166^Ho]holmium targeting bone and bone marrow lesions [111,112,113,114]. [^166^Ho]holmium is an inexpensive β¯-emitter, majorly used in the treatment of multiple myeloma [112,115].

[^166^Ho]Ho-phosphonates are taken up rapidly by bone and show [112,114,116] significant in-vitro stability after 72 h [112]. The free cation and the complex are excreted via kidneys [112]. One of the advantages of [^166^Ho]Ho-phosphonates is the negligible accumulation in extra-skeletal organs [112]. A dose of 1110 MBq has been used for therapy [114]. Also, administered activity less than 55,500 MBq estimated to be safe for kidneys to be used for bone marrow ablation in multiple myeloma patients [114]. Its potential efficacy in bone-pain palliation of multiple myeloma and bone metastases is being contemplated [112,116,117]. In a pilot study, [^166^Ho]Ho–DOTMP was used for the treatment of bone metastases in breast cancer patients showing median time to progression of 10.4 months [118]. In addition, owing to the remarkably higher radiation dose to the bone marrow, combination of [^166^Dy]dysprosium/[^166^Ho]Ho-EDTMP is proposed as an agent for bone marrow ablation [113].

Finally, the successful experience with radioisotopes labeled with phosphonate in addition to physical characteristics of [^166^Ho]Ho maybe encouraging to perform further studies investigating the efficacy of [^166^Ho]Ho-phosphonates in bone-pain palliation/treatment of multiple metastases or multiple myeloma. In addition, its potential in bone marrow ablation is intriguing. However, far more studies are needed before these investigational radiopharmaceuticals become a part of the clinical practice.

## 8. [223. Ra]radium-Dichloride ([^223^Ra]RaCl)

Similar to [^89^Sr]strontium, [^223^Ra]radium is a calcium mimetic isotope and deposits on hydroxyapatite [12]. [^223^Ra]Ra with a half-life of 11.4 days decays via a cascade of α, β¯ and γ emissions [12]. Ninety-four percent of the total decay energy is released from α-particles [12].

The approved injection dose is 55 KBq/kg body weight every month for a total of 6 cycles [119]. The excretion is mainly through gastrointestinal system, raising concern in the co-existence of constipation and inflammation in bowels [35].

Given the substantially shorter range of [^223^Ra]Ra in tissues, hematological toxicity is expected to be less in comparison with β¯-emitters [36]. Mild and reversible myelosuppression occurs with a nadir in 2–4 weeks after intravenous injection resolving in 6 weeks after administration [10,20]. In addition, diarrhea, nausea and vomiting occur in ≥ 10% of cases [35].

[^223^Ra]RaCl has been primarily evaluated for pain palliation of bone metastases, which has demonstrated a reduction of bone pain in 40–71% of patients in one study and 29–75% in another, showing a trend of higher response rates with the currently approved dose [37,38]. Also, significantly improved quality of life and higher pain relief, as well as longer time to first symptomatic SRE, need for EBRT and initial opioid use have been demonstrated [120,121] [35,121,122]. The typical time to decrease in pain intensity and duration of pain palliation is 1–8 and 6 weeks, respectively [20]. [^223^Ra]RaCl results in pain relief only in a portion of patients; however, the completion of all 6 courses of the therapy is indicated for achieving OS benefit in most cases [123]. Discontinuation of [^223^Ra]RaCl therapy should be considered in cases of significant hematologic derangements as well as evidence of rapid disease progression and visceral metastases [124].

Recently, the probability of benefits of [^223^Ra]RaCl administration in earlier stages of bone metastases in mCRPC has been proposed showing that patients with lower numbers of bone metastases have better outcome [125]. Metastases to the axial skeleton and the value of alkaline phosphatase were reported as significant predictors of the survival [125]. In addition, in a study, asymptomatic patients with two or more skeletal lesions have revealed more favorable results, demonstrating lower adverse events [126]. These findings may provoke further studies using [^223^Ra]RaCl at initial diagnosis of bone metastases.

Evaluation of response to therapy for bone metastases is challenging. This is not readily possible with CT or bone scintigraphy. Although PET/CT imaging using bone-seeking and tumor-specific tracers are not yet approved for this indication, the published studies, as well as our clinical experience revealed promising role of PET/CT in the assessment of treatment response (Figure 1 and Figure 2) [7,127]. Also, recent studies have shown that diffusion-weighted MRI may be of value in this setting [128]. Other traditional measures, like a decline in PSA level, are not reliable means to assess response following therapy with [^223^Ra]RaCl [127].

Although it is well-known that visceral metastases preclude patients from participation in [^223^Ra]RaCl therapy, an investigator has a contradictory view and propose that although survival benefit may be lower in these group of patients, it does not necessarily translate to poor response to therapy [129].

Therapy of mCRPC includes different strategies involving androgen receptor inhibitors, EBRT, chemotherapy (docetaxel) and immunotherapy [130]. Androgen receptor inhibitors are the main modality of treatment in mCRPC patients. However, patients can become resistant to both enzalutamide and abiraterone [127,131]. Considering the different mechanism of action, [^223^Ra]RaCl can become a qualified alternative to the common practice, switching to another androgen receptor inhibitor in cases of drug resistance [13].

The combination of hormonal therapy and [^223^Ra]RaCl is the subject of ongoing clinical studies [130]. [^223^Ra]RaCl has been used in combination with either abiraterone or enzalutamide; however, its impact on OS is still under investigation [132,133]. Preliminary data have demonstrated increase in survival with combination therapy. These studies are still ongoing to establish the impact on survival. Noteworthy, an improvement in event-free survival in mCRPC patients treated with abiraterone, corticosteroids and [^223^Ra]RaCl has not been achieved [134]. Interestingly, in the combination group, an increased risk of fracture has been reported, compared to the placebo arm [134].

Moreover, EBRT has been used safely along with [^223^Ra]RaCl [127]. EBRT prior to [^223^Ra]RaCl administration has not shown an association with compromised hematologic reserve [135]. The different mechanism of action of these two modalities advocates more studies on the combination therapy to evaluate the impact on the outcome.

Additionally, bone-pain palliation with denosumab or bisphosphonate in combination with [^223^Ra]RaCl has been investigated. Increase in OS has been observed only with denosumab [132]. In addition, the treatment with [^223^Ra]RaCl plus bisphosphonates has prolonged time to symptomatic SRE [136].

The safety of docetaxel administration prior to [^223^Ra]RaCl therapy has previously been studied [137]. Although this sequence of therapy seems to be well-tolerated, there has been an increased incidence of grade III/IV thrombocytopenia as well as a risk of decreased time to SRE, in the [^223^Ra]RaCl group [137]. However, [^223^Ra]RaCl has increased survival, regardless of previous docetaxel treatment [137]. In addition, the concomitant treatment with docetaxel and [^223^Ra]RaCl, compared to docetaxel, has demonstrated a favorable antitumor activity has prolonged biochemical suppression [138] and time to progression [139].

The choice between the sequence of [^223^Ra]RaCl and docetaxel therapy is mainly based on the involved organs. The absence of visceral involvement favors the use of [^223^Ra]RaCl as the primary agent [13]. Further larger studies on the risks and benefits of concurrent administration of chemotherapy and [^223^Ra]RaCl are required. The same is true for co-administration of β¯-emitting bone-targeted therapies and [^223^Ra]RaCl, as their safety profile needs further investigation [127].

Other newer agents such as programmed cell death (PD-1) inhibitors have also been approved for types of mCRPC that are microsatellite instability (MSI)-high and DNA mismatch repair-deficient [130]. The combination of these agents (atezolizumab) with [^223^Ra]RaCl in mCRPC with bone metastases is currently under investigation [130]. Other newer agents include DNA damage response inhibitors, such as Olaparib, that has revealed a favorable response rate in mCRPC patients, in a phase II trial. Ongoing trials are also investigating the effect of these newer agents in association with [^223^Ra]RaCl [130].

Not surprisingly, [^223^Ra]RaCl is the subject of ongoing trials in other bone dominant metastatic cancers such as breast [140,141] and lung cancers [142]. In a phase II trial, [^223^Ra]RaCl has been administered with different hormonal therapy agents [141]. The primary end-point was disease control at 9 months, achieved in 49% of patients [141]. In addition, the study evaluating the efficacy of combined therapy of [^223^Ra]RaCl in association with hormonal therapy and a chemotherapy drug (capecitabine) is still in progress [140].

Ultimately, the studies are in progress, evaluating the further benefits of [^223^Ra]RaCl. [^223^Ra]RaCl seems a promising therapeutic agent beside its palliative effects, used alone or in combination with other treatments or palliative agents in mCRPC patients and it may find its way in other bone dominant metastatic malignancies.

## 9. Discussion

Bone metastases develop in approximately 30% of cancer patients and occur in up to 60–90% in late phases [26]. Therapeutic approaches have been extensively investigated to reach for an optimum method. Since the first introduction of radionuclide therapy in bone-pain palliation in the 1940 s, numerable radioisotopes and pharmaceuticals have been exploited to target the bone lesions and alleviate the pain. Each demonstrates particular characteristics, benefits and drawbacks (Figure 3). However, the choice extremely rests on the logistic feasibility of the tracer and then on the patient’s status (renal function, bone marrow reserve), cancer extent (extraskeletal lesions, the bulk of the tumor) and physical properties of radioisotopes (Figure 4).

No significant superiority in pain reduction has been documented using different β¯-emitters [18]. The α-emitter, [^223^Ra]RaCl, also shows a response rate of 50–60% [146], not beyond the reported range for β¯-emitters. However, the effects on extending survival parameters make [^223^Ra]RaCl a more appealing agent to be employed.

Noteworthy, patients must comply with requirements for radionuclide treatment, that is, to have osteoblastic activity in symptomatic bone metastases, sufficient renal function, bone marrow reserve and life expectancy [147]. In addition, presence of spinal cord compression, high risk of fracture in weight-bearing bones, pregnancy and breastfeeding are contraindication for radionuclide therapy [3,147].

### 9.1. Concomitant Extraskeletal Involvement

When the disease is not confined to the skeleton, other than exclusively bone-seeking agents are mandatory to target all tumoral lesions. Disease-specific radiopharmaceuticals (including [^177^Lu]Lu–PSMA in prostate cancer, [^177^Lu]Lu/[^90^Y]Y-conjugated peptides in neuroendocrine tumors, [^131^I]Iodine in differentiated thyroid cancer, [^131^I]I-MIBG in pheochromocytomas/paragangliomas, etc.) provide simultaneous radiation to the skeletal and visceral lesions. In addition, bone-seeking agents may be administered when the disease-specific radiopharmaceuticals fail to reduce pain, considerably. This hypothetical application needs trials to depict safety and efficacy.

### 9.2. Life Expectancy

Patients with very short life expectancy may not be optimal candidates for bone-pain palliation with radiopharmaceuticals since the response to therapy commonly commences with a delay of 1–4 weeks with reports of exacerbation of pain (flare-response) in a few first days [3]. If patients with shorter predicted life expectancy are to receive this therapy, radioisotopes with shorter half-lives would be better options, such as [^188^Re]rhenium and [^153^Sm]samarium, as well as [^186^Re]rhenium and maybe [^166^Ho]holmium. In contrast, in case of predicted long life expectancy, other than [^223^Ra]radium and [^89^Sr]strontium revealing longer palliation effects may be proper options. In addition, repeated doses of other radiopharmaceuticals or combination with other therapies may be reasonable.

### 9.3. Bone Marrow Status

Bone marrow reserve can generally be the dose-limiting factor. Before administration of bone-seeking agents, hematological indices typically should meet the values of hemoglobin ≥90 g/L, WBC ≥3 × 10^9^/L and platelet ≥100 × 10^9^/L in most of the cases [3]. These indices are as follows for [^223^Ra]RaCl: hemoglobin ≥10 g/L, absolute neutrophil count ≥1.5 × 10^9^/L and platelet ≥100 × 10^9^/L [35]. In patients with compromised bone marrow reserve, radiopharmaceuticals with less hematological toxicities or shorter particle range are favored, including [^153^Sm]samarium and [^177^Lu]lutetium, as well as [^223^Ra]radium.

### 9.4. Renal Function

The clearance of radiopharmaceuticals excreted via kidneys is prolonged in renal insufficiency, increasing the whole-body dose and myelosuppressive side effect [3]. Hence, in case of mild to moderate decreased renal function, [^223^Ra]radium (with negligible renal excretion) and β¯-emitters with a shorter half-life and with reduced doses ([^153^Sm]samarium, [^186^Re]rhenium and maybe [^177^Lu]lutetium, [^188^Re]rhenium or [^166^Ho]holmium) are favored.

### 9.5. Tumor Size

The size of the tumoral lesions can also play a role in opting the proper radiopharmaceutical. The energy of the emitted β¯-particle is proportional to the range of penetration in the tissue. Hence, radiopharmaceuticals with higher β¯-particle energy, such as [^188^Re]rhenium and [^166^Ho]holmium can be more effective in larger lesions than small ones. As a rule of thumb, [^177^Lu]lutetium and [^153^Sm]samarium can be used in small-sized lesions. Moreover, α-particles deliver substantially high energy in a short distance and can be employed in small-sized metastases, as well.

Historically, regarding the merely palliative effects of traditional β¯-emitters, radiopharmaceuticals become a part of patient treatment when physicians are at the end of their rope and the life expectancy is poor. However, this approach may change in the guidelines, in the future, regarding: first, β¯-emitters are more effective when they are used in early stages and limited skeletal involvement and when the patient’s performance state is higher [74,85]; second, the combination therapy is being more evaluated and demonstrate promising results. Other than chemo- or radiotherapy, combination with bisphosphonates or denosumab, a RANK ligand inhibitor, are barley investigated; third, it is well-known that α-emitters are more toxic for tumoral cells and show significant survival benefits.

Future directions are concisely mentioned for each radiopharmaceutical in the pertinent section. On the whole, further studies evaluating the role of radiopharmaceuticals in the earlier course of the disease and the combination with other therapies may clarify the role of nuclear medicine in the palliation or treatment of bone metastases.

## 10. Conclusions

In summary, the management of painful bone metastases is still a challenging issue. Bone-pain palliation with β¯-emitting isotopes is being conducted for decades, revealing acceptable results. Several radiopharmaceuticals have been and still are being investigated to find an optimum agent. Based on the current data, each therapeutic radioligand has its own advantages and limitations without any substantial superiority over others. The impact of radionuclide therapy on overall and disease-free survival as well as SRE has been shown in the published studies. However, further investigations with larger patient populations are needed to confirm their prognostic roles. Nevertheless, the introduction of [^223^Ra]radium with its advantageous impact on survival cast a shadow over other radiopharmaceuticals; however, unfortunately, the expensive [^223^Ra]radium is not globally available. Combination treatments using bone-seeking radiopharmaceuticals with other therapies have demonstrated safety and favorable results in some studies. Hence, further investigations are mandatory to establish the role of bone-seeking and tumor-targeted radionuclide therapy, optimal timing and combined protocols, in the palliation or treatment of bone metastases.

## Figures and Tables

**Figure 1 jcm-09-02622-f001:**
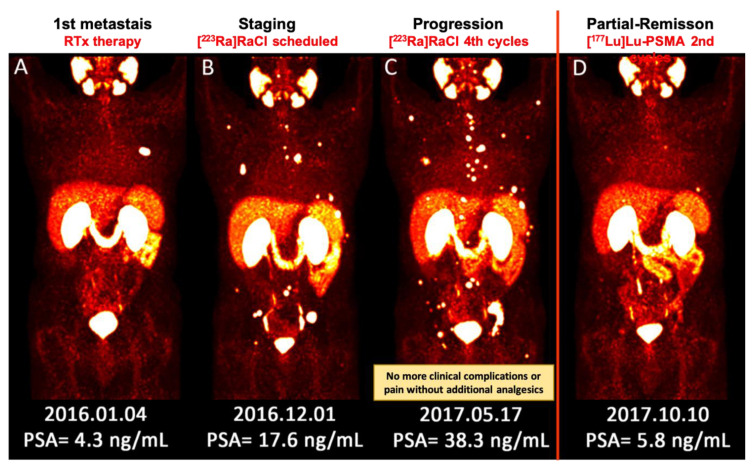
(**A**–**D**) Sequential images (MIP) of [^68^Ga]Ga–PSMA PET/CT scan from a patient with metastatic prostate cancer. (**A**) He underwent local radiotherapy for single bone metastasis in the left 6^th^ rib; (**B**) After approximately one year, he developed multiple painful bone metastases. He received 4 cycles of [^223^Ra]RaCl therapy. Pain reduced significantly and he no longer needed analgesic medication; (**C**) Nevertheless, the PSA level was apparently rising and the disease was progressing. The therapeutic plan was switched to [^177^Lu]Lu–PSMA. Afterwards, the response to therapy was clearly captured by [^68^Ga]Ga–PSMA PET/CT after 2 cycles of [^177^Lu]Lu–PSMA therapy, which was concomitant with the significant biochemical response (decrease in PSA level). [^177^Lu]Lu–PSMA: [^177^lutetium]Lu-prostate-specific membrane antigen; [^223^Ra]RaCl—[^223^radium]ra-dichloride; [^68^Ga]Ga–PSMA PET/CT—[^68^gallium] Ga-prostate-specific membrane antigen positron emission tomography/computed tomography; MIP—maximum intensity projection; PSA—prostate-specific antigen; RTx—radiotherapy.

**Figure 2 jcm-09-02622-f002:**
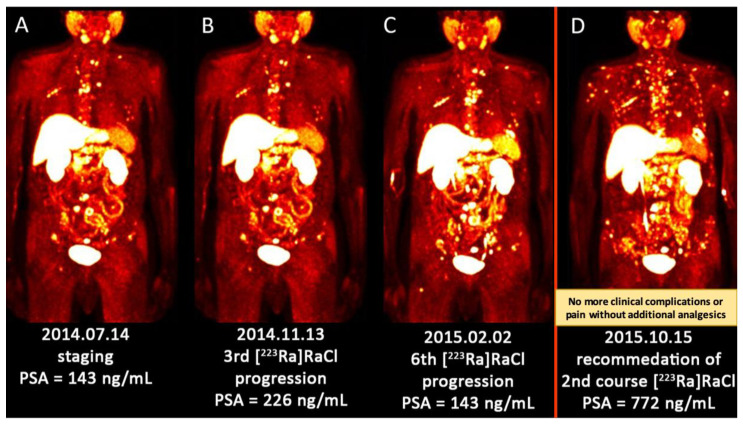
(**A**–**D**) Sequential images (MIP) of [^18^F]FCH PET/CT scan from a patient with multiple metastatic prostate cancer. (**A**) There were multiple bone metastases predominantly in the axial skeleton. He received 3 cycles of [^223^Ra]RaCl; (B) The PSA was increasing (more than 150%) after 3 cycles of [^223^Ra]RaCl. He finished 6 cycles of [^223^Ra]RaCl therapy. He became pain-free and ceased to take analgesic medication; (**C**) The metabolism imaging showed rather stable metabolic disease although the PSA level was slightly increased; (**D**) After 8 months of follow-up, there was an aggressive anatomic, metabolic and biochemical progression of the disease. Regarding the previous and slow disease progression under [^223^Ra]RaCl, he was recommended to receive the 2nd course of [^223^Ra]RaCl treatment/palliation. [^18^F]FCH PET/CT—[^18^Fluorine] fluorocholine positron emission tomography/computed tomography; MIP—maximum intensity projection; PSA—prostate-specific antigen; [^223^Ra]RaCl—[^223^radium]ra-dichloride.

**Figure 3 jcm-09-02622-f003:**
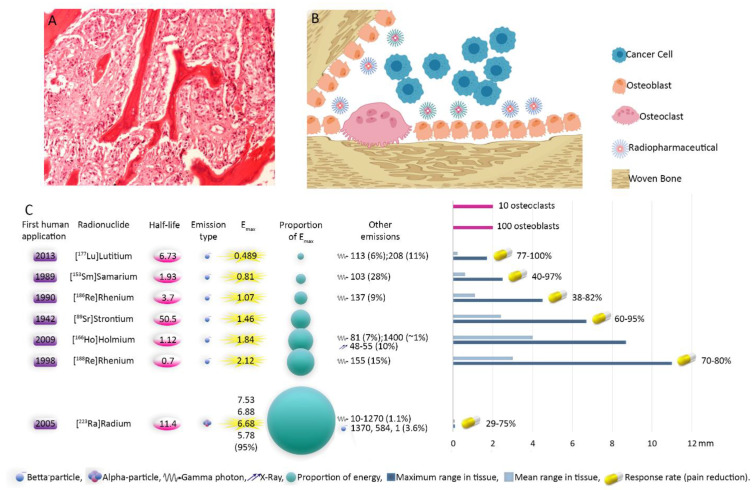
Illustration of bone metastases and physical characteristics of radiopharmaceuticals. Cancerous cells from prostate origin produce cytokines activating osteoblasts. On the other hand, their homing is facilitated by secreted factors from osteoblasts. In addition, chemoattractants created by newly formed bone matrix increase the invasiveness of cancer cells. This cycle continues destructing the bone. Bone-seeking radiopharmaceuticals reside in regions with bone turn over. Accumulation of radiopharmaceuticals in the osteoblastically active regions, which is induced or increased significantly by tumoral cells infiltration, leads in mainly irradiation of metastatic sites with only minimal impact on the normal tissues. Eventually, the tumoral tissues shrink, osteoblastic activity is inhibited and stimulation of the periosteal pain receptors is reduced [10,31]. (**A**) Histopathology image of bone metastasis; (**B**) schematic illustration of bone metastasis with predominant osteoblastic proliferation. Note the accumulation of radiopharmaceuticals in areas with osteoblastic activity. Depending on the range of the agent, the cancerous and normal cells will be affected; (**C**) physical characteristics and efficiency of radiopharmaceuticals. The response rates (Table 1) pertain to bone-seeking pharmaceuticals labeled with the mentioned radionuclides. E_max_: Maximum particle energy. First human application [21,89,118,143,144,145].

**Figure 4 jcm-09-02622-f004:**
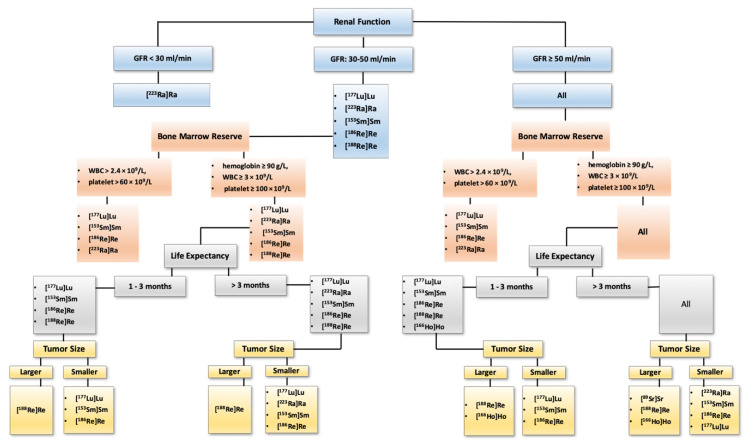
A proposed algorithm for selecting an appropriate radiopharmaceutical for bone-pain palliation therapy. Bone-pain palliation with radiopharmaceuticals is offered when the metastatic sites are widespread or not amenable to the local therapy. Prostate cancer with bone metastases accompanied with minimal lymph node metastases should be treated with [^223^Ra]RaCl. Those with concomitant visceral metastases should be considered for treatment with [^177^Lu]Lu–PSMA. Metastatic thyroid cancer, neuroendocrine tumors, pheochromocytomas and paragangliomas must be treated with associated targeted radiopharmaceuticals, namely [131I]Iodine, [^177^Lu]Lu/[^90^Y]Y-conjugated peptides and [^131^I]I-MIBG, respectively. For other malignancies with painful bone metastases the above algorithm is proposed. Noteworthy, only widely established agents are [^223^Ra]RaCl (for prostate cancer), [^153^Sm]Sm–EDTMP and [^89^Sr]SrCl. In conjunction with radiopharmaceuticals, other hormonal therapies (e.g., androgen deprivation therapy for prostate cancer, octreotide for neuroendocrine tumors, etc.) should be continued. Besides radiopharmaceuticals, concomitant use of bisphosphonates and denosumab, as well as other therapies should be considered in the context of trials. This proposed general approach is based on our personal experience; however, decision making should be based on local experiences, availability of the radioligands, national regulations and guidelines as well as reimbursement provided by health systems.

**Table 1 jcm-09-02622-t001:** Physical characteristics, efficacy and toxicity of radiopharmaceuticals.

Radioisotope	Pharmaceutical	Half-Life (Days)	Emission	β Energy: Maximum/Mean (Mev)	Other Emissions (KeV) Abundance (%)	Range in Soft-Tissue: Maximum/Mean Maximum in Bone (mm)	Excretion (Main)	Administration Dose (MBq)	Response Rate (%)	Response Duration (Months)	Distinct Feature	References
[^89^Sr]strontium	Dichloride	50.5	β¯	1.46/0.583	–	6.7/2.4/3.0	Renal	150	60–95	3–6.5	Longer typical duration of response	[4,8,9,19,20,21]
[^153^Sm]samarium	EDTMP	1.93	β¯^-^	0.81/0.233/	γ: 103 (28%)	2.5/0.6/-	Renal	37 MBq/kg	40–97	2–3	Widely investigate and available	[10,20,22,23,24,25]
[^186^Re]rhenium	HEDP	3.7	β¯ γ	1.07/0.349	γ: 137 (9%)	4.5/1.1/-	Renal	1295	38–82	5–12	–	[17,26]
[^188^Re]rhenium	HEDP	0.7	β¯^-^γ	2.12/0.64	γ: 155 (15%)	11/3/-	Renal	1100–3300	70–80	3–6	Potential antitumor effect	[18,27,28,29,30]
[^177^Lu]lutetium	EDTMP DOTMP	6.73	β¯^-^γ	0.489/0.133	γ: 113 (6%), 208 (11%)	1.7/0.23/-	Renal	1295–2590	77–100	1–4	Widely available	[10,21,31]
[^166^Ho]holmium	DOTMP EDTMP	1.12	β¯ γ x	1.84/0.67	γ: 81 (7%), 1400 (~1%) x: 48 to 55 (10%)	8.7/4.0/3.8	Renal	1110/<55,500	–	–	Efficacy in bone marrow ablation	[16,31,32,33,34]
[^223^Ra]radium	Dichloride	11.4	α β¯ γ	Maximum α energy: 7.53, 6.88, 6.68, 5.78 (95.3%)	β^-:^ 1370, 584, 1 (3.6%) γ: 10–1270 (1.1%)	<0.1/0.05–0.08/-	Gastrointestinal	55 KBq/kg × 6	29–75	1.5	Therapeutic effect	[10,35,36,37,38]

DOTMP—1,4,7,10-tetraazacyclododecane-1,4,7,10-tetramethylene phosphonate; EDTMP—ethylene diamine tetramethylene phosphonate; HEDP—hydroxyethylidene diphosphonate.

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
