# Peer review of "Targeted Palliative Radionuclide Therapy for Metastatic Bone Pain"

_jcm, 2020, doi:10.3390/jcm9082622_

Round 1
Reviewer 1 Report
A nice, comprehensive review of the subject. I'd just suggest changing the figures 1 and 2, so that the panels are A-D, rather than D-A.
Author Response
I'd just suggest changing the figures 1 and 2, so that the panels are A-D, rather than D-A.
Response: We appreciate the reviewer’s comment. The figures are revised.
Reviewer 2 Report
This review provide an overview of bone-seeking radiopharamaceuticals used for bone pain palliation, their effectiveness and toxicity. This document is a good comprehensive review of existing radiopharmaceuticals.
The subject of this article is very current and interesting. Besides, the manuscript is well written and very clear.
It’s a pity that the studies reviewed only too rarely include quality of life items.
Author Response
It’s a pity that the studies reviewed only too rarely include quality of life items.
Response: We genuinely appreciate the reviewer’s comment on enhancing our work. The reduction in pain is generally associates with improvement in quality of life. The impact on quality of life has been mentioned in each part very briefly. Moreover, two sentences are added to the manuscript and articles to the references.
“Besides pain reduction, significant improvement of quality of life has been reported [44, 45]. However, the improvement in quality of life generally follows pain reduction [46].”
(Section: Strontium-dichloride, page 8, 4th paragraph)
Reviewer 3 Report
This review is well written and structured.
The main elements of interest regarding this topic are discussed.
The chapters are detailed and collects relevant data from the literature.
I suggest to perform some minor revision.
- Considering the multitude of therapeutic options in bony metastatic tumor add some treatment flow-chart could be useful to lead the choice of appropriate drug.
- In section 2 dedicated to Strontium –dichloride describing the combination therapy with Zoledronic Acid, the authors could add the follow reference “ G. Storto et al. Combined therapy of Sr-89 and zoledronic acid in patients with painful bone metastases Bone . 2006 Jul;39(1):35-41”
- In section 8 dedicated to Radium-dichloride I would avoid the sentence “[223-Ra] RaCl can be safely used in combination with either abiraterone or enzalutamide" particularly considering the association with Ra-223+Abi and the results of the ERA-trial.
Author Response
- Considering the multitude of therapeutic options in bony metastatic tumor add some treatment flow-chart could be useful to lead the choice of appropriate drug.
Response: We appreciate the reviewer’s comment. It is added as a figure to the discussion part.
(Section: Discussion, page 24, figure 4)
- In section 2 dedicated to Strontium –dichloride describing the combination therapy with Zoledronic Acid, the authors could add the follow reference “ G. Storto et al. Combined therapy of Sr-89 and zoledronic acid in patients with painful bone metastases Bone . 2006 Jul;39(1):35-41”
Response: Thanks for the comment. The article is added to the manuscript and references.
“Finally, the combination therapy of [89Sr]SrCl and Zoledronic Acid for bone metastases has shown superiority in terms of reduction of bone pain, analgesic drug use, and time to decrease in pain, as well as improvement of the quality of life compared to [89Sr]SrCl- or Zoledronic Acid-alone [48]. There has been no significant higher rate of toxicity in the combined method [48].”
(Section: Strontium-dichloride, page 9, 2nd paragraph)
- In section 8 dedicated to Radium-dichloride I would avoid the sentence “[223-Ra] RaCl can be safely used in combination with either abiraterone or enzalutamide" particularly considering the association with Ra-223+Abi and the results of the ERA-trial.
Response: We genuinely appreciate the reviewer’s comment on enhancing our work. We meant it is safe considering the hematological toxicity. It was vague. It is revised.
(Section: Radium-dichloride, page 16, 3rd paragraph)